# The Effect of (1*S*,2*R*-((3-bromophenethyl)amino)-*N*-(4-chloro-2-trifluoromethylphenyl) cyclohexane-1-sulfonamide) on *Botrytis cinerea* through the Membrane Damage Mechanism

**DOI:** 10.3390/molecules25010094

**Published:** 2019-12-25

**Authors:** Jingnan Peng, Kai Wang, Tingyue Feng, Huazhong Zhang, Xinghai Li, Zhiqiu Qi

**Affiliations:** Department of Pesticide Science, Plant Protection College, Shenyang Agricultural University, Shenyang 110866, Liaoning, China; pjn941106@163.com (J.P.); wangkai12@syau.edu.cn (K.W.); SunnyFTY@outlook.com (T.F.); zhz961116@163.com (H.Z.); xinghai30@163.com (X.L.)

**Keywords:** SYAUP-CN-26, *Botrytis cinerea*, membrane damage

## Abstract

In recent years, *Botrytis cinerea* has led to serious yield losses because of its resistance to fungicides. Many sulfonamides with improved properties have been used. (1*S*,2*R*-((3-bromophenethyl)amino)-*N*-(4-chloro-2-trifluoromethylphenyl)cyclohexane-1-sulfonamide) (abbreviation: SYAUP-CN-26) is a new sulfonamide compound that has excellent activity against *B. cinerea*. This study investigated the effect of SYAUP-CN-26 on electric conductivity, nucleic acids leakage, malondialdehyde (MDA) content, and reducing sugars and membrane structure reduction of *B. cinerea.* The results showed that the cell membrane permeability of *B. cinerea* increased with increasing concentrations of SYAUP-CN-26; meanwhile, the sugar content decreased, the malondialdehyde content increased, and relative electric conductivity and nucleic acid substance leakage were observed in the cell after exposure to 19.263 mg/L SYAUP-CN-26 for 24 h. After 48 h of exposure to 1.823 mg/L and 19.263 mg/L SYAUP-CN-26, the cell membranes of *B. cinerea* mycelia were observed to be damaged under propidium iodide (PI) and transmission electron microscopy (TEM) observations. It is assumed that SYAUP-CN-26 was responsible for the damage of cell membrane. Overall, the results indicate that SYAUP-CN-26 could inhibit the growth of *B. cinerea* cells by damaging the cell membranes.

## 1. Introduction

As a result of their diversity and wide distribution, plant-pathogenic fungi cause economic losses and threaten food security [1,2,3]. *Botrytis cinerea,* with a broad host range and worldwide distribution, is one of the most aggressive phytopathogens and is responsible for significant yield losses [4]. It can infect almost all parts of the plant, including fruits, flowers, stems and leaves, which greatly reduces plant yield and quality [5,6]. *B. cinerea* thrives at the highly humidity condition and the optimum temperature for growth is around 20 °C [7]. Plant tissue can be quickly eroded by *B. cinerea*, producing a large number of asexual spores, which can be transmitted by wind or insects [8]. Although there are many fungicides that can control it, because of the fast development of its resistance, many fungicides have failed [9,10,11,12,13], such as quinone outside inhibitors, benzimidazole, carbamates, phenylpyrroles, dicarboximides, anilinepyrimidines and hydroxyanilide fungicides [14,15,16]. Therefore, it is important to develop new kinds of fungicides with new structures and mechanisms.

Sulfonamides were the first medicines used against bacteria [17]; they can also act as herbicides for controlling weeds [18]. However, there are currently only four sulfonamide fungicides: flusulfamide, which acts against *Plasmodiophora brassicae* [19], tolnifanide, which acts against *Cladosporium cucumerinum*, and cyazofamid and amisulbrom, which act against *Oomycetes* [20,21]. Therefore, more attention should be paid to sulfonamides in the development of fungicides with improved properties [22,23,24] and the ability to target a wider array of pathogens. Recently, some sulfonamides, such as cycloalkylsulfonamides and benzenesulfonamides, were reported as being able to control *Sclerotinia sclerotiorum*, *Fusarium oxysporum*, *Thanatephorus cucumeris*, and *B. cinerea* [25,26]. In addition, their derivatives, such as 1,3,4-thiadiazoles, coumarins, pyrans, chesulfamide, and 2-amino-6-oxocyclohexenylsulfonamides, were reported to be able to control *B. cinerea*, *Corynespora cassiicola,* and *Cladosporium cucumerinum* [27,28] with their antibacterial spectrum being obviously different from other sulfonamide fungicides.

As shown in Figure 1, (1*S*,2*R*-((3-bromophenethyl)amino)-*N*-(4-chloro-2-trifluoromethylphenyl) cyclohexane-1-sulfonamide) (abbreviation: SYAUP-CN-26) is a cycloalkyl compound containing one sulfonyl group, which is synthesized using the lead compound chesulfonamide. In a previous study, chesulfonamide was shown to effectively control *B. cinerea* and *C. cassiicola* with a strong preventive activity [27]. SYAUP-CN-26 is a derivative of chesulfonamide, which has a better activity against *Botrytis cinerea* than chesulfnamide [29]. Until now, there have only been a few published studies focusing on these compounds and only biological assays have been tested on SYAUP-CN-26 and other derivative, such as, inhibitory effect on mycelium growth and spore germination of *B. cinerea*, fungicidal activities in pot tests and control efficacy in field [30]. It is necessary to clarify the mechanism of action of SYAUP-CN-26 on *B. cinerea*. It is hoped that this research will provide a theoretical basis for further optimization of such compounds and the development of new fungicides.

This study assessed the electric conductivity, nucleic acid leakage, reductions in sugar and malondialdehyde (MDA) content, morphology and ultrastructural observation to uncover the action mechanism of SYAUP-CN-26 microscopically and physiologically.

## 2. Results

### 2.1. Relative Electric Conductivity of B. Cinerea

There was no obvious difference between the acetone control and blank control for all indicators in the study of the relative electric conductivity, nucleic acid, malondialdehyde contents, reducing sugar content; for this reason, only the blank control was used for the results and discussion.

As shown in Figure 2, the relative electric conductivity value for each treatment increased with time. After incubation with 1.823 mg/L, 19.263 mg/L, and a minimum inhibitory concentration (MIC, 79.754 mg/L) of SYAUP-CN-26 for 2 h, the values were 1.0033, 1.0026, and 1.0058 times that of the control, respectively. After 24 h, the relative electric conductivity values for 1.823 mg/L, 19.263 mg/L and 79.754 mg/L of SYAUP-CN-26 were 1.02, 1.04, and 1.05 times of the control, respectively; these were significantly higher than the control (*p* < 0.05).

### 2.2. Nucleic Acid of B. Cinerea

As shown in Figure 3, after incubation with the 79.754 mg/L of SYAUP-CN-26 for 2 h, the values of nucleic acid were 1.01 times that of the control. After 24 h, the relative electric conductivity for 1.823 mg/L, 19.263 mg/L, and 79.754 mg/L of SYAUP-CN-26 were 1.02, 1.03, and 1.04 times that of the control, respectively; these were significantly higher than the control (*p* < 0.05).

### 2.3. Malondialdehyde (MDA) Contents of B. Cinerea

The malondialdehyde (MDA) content can reflect the degree of membrane lipid peroxidation, which is one of the important indicators of membrane damage [31]. Once the cell membrane was damaged, it would be released into the extracellular space. The degree of the damage to membrane system can be indirectly evaluated by detecting malondialdehyde (MDA) content [32]. The results show that as the concentrations of SYAUP-CN-26 increased, the malondialdehyde (MDA) content also increased (Figure 4). The MDA content of *B. cinerea* treated with 1.823 mg/L and 19.263 mg/L of SYAUP-CN-26 for 24 h were 1.24 and 1.38 times that of the control, respectively. The MDA of *B. cinerea* treated with 79.754mg/L was 2.30 times that of the control, which is significantly higher (*p* < 0.05).

### 2.4. Reducing Sugar Content of B. Cinerea

In general, the reducing sugar content continuously decreased for *B. cinerea* treated with 1.823 mg/L, 19.263 mg/L, and 79.754 mg/L (Figure 5). The reductions in the reducing sugars content of *B. cinerea* incubated for 2 h, 6 h, and 12 h were slightly lower than those from the control. After 24 h, the reductions in the reducing sugar contents reached 93.88% and 90.18% for 1.823 mg/L and 19.263 mg/L, respectively. Regarding the 79.754mg/L-treated group, the reduction in reducing sugars content was 74.00%; that of the control was significantly lower (*p* < 0.05).

### 2.5. Membrane Integrity of the *B. Cinerea* Spore

This study used propidium iodide (PI) to determine whether SYAUP-CN-26 damaged the membrane’s integrity in *B. cinerea* spores. According to Figure 6b,c (Yellow arrow), most spores showed red fluorescence at 1.823 mg/L, and the spores all showed strong red fluorescence at 19.263 mg/L SYAUP-CN-26, indicating that the plasmatic membrane of the *B. cinerea* spores was markedly damaged. In addition, the spores did not present red fluorescence at 0 mg/L, indicating that the integrity of the membrane remained unaltered in the untreated spores (Figure 6a).

### 2.6. Morphology and Ultrastructure of *B. Cinerea*

As shown in Figure 7, after treatment with 0 mg/L SYUAP-CN-26, the mycelia presented a typical morphology with linear and homogeneous mycelia: the mycelium surface was smooth, regular, and complete (Figure 7A). The spores were of a normal, oval, plump, and homogenous morphology (Figure 7a). However, after treatment with 1.823 mg/L and 19.263 mg/L SYAUP-CN-26, the mycelia and spores showed irregular shrinkage and cell malformation, with obvious collapse and destruction being visible on the surface of mycelia (Figure 7B,b,C,c, yellow oval label).

As shown in Figure 8A, after treatment with 0 mg/L SYUAP-CN-26, the cell membranes of *B. cinerea* mycelia were uniform, the cytoplasmic matrixes were abundant, and the structure of main cytoplasmic organelles was normal. In contrast, these normal morphological structures conspicuously changed after treatment with SYAUP-CN-26. The ultrastructural changes observed included the cell walls and cell membranes becoming sparse (Figure 8B(I)) with 1.823 mg/L SYAUP-CN-26. The cell ultrastructure damage was more serious, with the cell walls and cell membranes rupturing (Figure 8C(III)) and cytoplasmic organelles decreasing and vacuolating at 19.263 mg/L SYAUP-CN-26 (Figure 8C(II)).

## 3. Discussion

Sulfonamide compounds have always been of great interest to researchers because of their broad biological activity [26,33,34]. SYAUP-CN-26 is one cycloalkyl compound that contains a sulfonyl group, and there is no available research regarding its mechanism of action on phytopathogens.

Maintaining ion homeostasis plays an important role in preserving the cellular energy status and membrane coupling, which involves solute transport, metabolic control, and motion management. Therefore, even relatively small changes in the cell membrane can have a detrimental effect on cell metabolism and lead to cell death [35,36]. In this study (Figure 2), the results show that the relative electric conductivity was significantly inhibited by 1.823 mg/L, 19.263 mg/L, and the 79.754 mg/L of SYAUP-CN-26 after 24 h, which means that the membrane permeability injuries caused by SYAUP-CN-26 led to leakage of ions. In addition, the results were similar to previous research, which found that the isoliquiritin acts against *Peronophythora litchi* through membrane damage [37].

MDA is one of the products of membrane lipid peroxidation, and so can reflect the degree of membrane lipid peroxidation, which is one of the important indicators of membrane damage [31]. Once the cell membrane is damaged, it is released into the extracellular environment. Therefore, the degree of damage to the membrane system can be indirectly evaluated by detecting the MDA content [32]. The results show that the MDA concentration of *B. cinerea* treated with 79.754 mg/L was significantly higher than the control after 24 h, which indicates that SYAUP-CN-26 induced plasma membrane peroxidation of *B. cinerea*, damaging its membrane system. Because of cell membrane damage, the permeability of the cell membrane and the content of extracellular nucleic acid increased [38].

The reduction of the reducing sugars can cause sugar deficiency in the mycelium, induce stress-activated protein release, interrupt material metabolism and transport, and ultimately lead to hyphae growth inhibition [37,39]. The results (Figure 5) show that after treatment with 1.823 mg/L and 19.263 mg/L SYAUP-CN-26, the reduction of reducing sugars decreased slightly, but SYAUP-CN-26 significantly decreased the reducing sugars content for the 79.754 mg/L group, indicating that the reduction in the reducing sugars may be due to the increase in cell membrane permeability, or SYAUP-CN-26 inhibiting the metabolism of the reducing sugars.

The hypothesis that SYAUP-CN-26 could damage cell membrane integrity and the ultrastructure of *B. cinerea* was verified by fluorescence microscope images, SEM images, and TEM images (Figure 6, Figure 7 and Figure 8). Ma et al. deemed this phenomenon to be due to the increase in cell permeabilization, thus causing cell membrane stability to be disrupted [38]. The SEM images (Figure 7) clearly show that SYAUP-CN-26 can cause the mycelia of *B. cinerea* to shrink, which might be due to SYAUP-CN-26 increasing the permeability of the cell membrane and causing the leakage of the cellular content. TEM analysis demonstrated that the cell membrane of *B. cinerea* mycelia was severely damaged, and certain cytoplasmic organelles were degraded by SYAUP-CN-26, which is consistent with previous results [5]. Previous studies have documented that the degree of damage to the cell membrane, such as inhibition of membrane component biosynthesis and destruction of the membrane structure, is one common way to evaluate the antiseptic qualities of potential fungicides [40]. For example, the natural plant-derived compound thymol can induce cell membrane damage in *Fusarium graminearum* through disrupting ergosterol biosynthesis [36].

Thus, in this study, we tried to uncover the mechanism of action of SYAUP-CN-26 using both microscopic and physiological observations. The results suggest that SYAUP-CN-26 can damage the cell membrane structure, leading to leakage of the cell contents. This was different from the previously studied mechanism of action of sulfonamide, which affected substance synthesis [41] or connected to impairment of the ATP energy generation system [21]. Of course, further studies are needed to explore the exact mechanisms by which SYAUP-CN-26 affected the cell membrane structure.

## 4. Materials and Methods

### 4.1. Collection of Isolate and Chemicals

*B. cinerea* was originally isolated from diseased tomatoes, purified by a single spore isolation technique and identified based on cultural and morphological characteristics, and identified by Shenyang Agricultural University Pesticide Laboratory. SYAUP-CN-26, provided by the Plant Protection College, Shenyang Agricultural University, was dissolved in acetone; the stock solution was 1 × 10^4^ mg/L and was stored at 4 °C in the dark.

### 4.2. Measurement of Relative Electric Conductivity

The permeability of *B. cinerea* cell membranes was expressed in terms of their electric conductivity [33]. The spores of *B. cinerea* used for experiments were collected from seven-day-old cultures of fungi grown on a potato dextrose agar (PDA) medium. The spore suspension (5 × 10^5^ cfu/mL) was inoculated in 40 mL of potato dextrose broth (PDB) medium (25 °C, 120 r/min) for 36 h. The mycelium pellets were centrifuged at 5000 *g* for 10 min and washed three times with sterilized water, the mycelia pellets were resuspended in 50 mL of sterilized water. SYAUP-CN-26 was added to the above suspensions. The final concentrations were 1.823 mg/L, 19.263 mg/L, and 79.754 mg/L. The concentrations of 1.823 mg/L, 19.263 mg/L, and 79.754 mg/L were EC_50_, EC_90_, and the MIC of SYAUP-CN-26 against *B. cinerea*, respectively. Each assay contained three replicates for each concentration. The treated suspensions were incubated (25 °C, 120 r/min) for 2, 6, 12, and 24 h. The electric conductivity of the *B. cinerea* suspension was detected with a conductivity meter (DDS-11A; Shanghai Leici Instrument Inc., Shanghai, China). Results are expressed as the amount of relative electric conductivity (ms/cm).

### 4.3. Measurement of Nucleic Acids

Nucleic acids from the extracellular medium were analyzed as described by Lewis et al. [42] with modifications. The spores of *B. cinerea* used for experiments were collected from seven-day-old cultures of fungi grown on PDA medium. The spores’ suspension (5 × 10^5^ cfu/mL) was inoculated in 40 mL of PDB medium (25 °C, 120 r/min) for 36 h. The cultured *B. cinerea* mycelia were weighed at 0.5 g and then put into 10 mL sterile water; SYAUP-CN-26 was added to *B. cinerea* suspension, the final concentrations were 1.823, 19.263 mg/L and 79.754 mg/L, then *B. cinerea* suspension was incubated (25 °C, 120 r/min) for 2, 6, 12, and 24 h. The nucleic acids in each treatment were determined using absorbance microplate reader (SpectraMax190, Molecular Devices, USA). The detected wave length was 260 nm [43].

### 4.4. Lipid Peroxidation

Malondialdehyde (MDA) was analyzed according to Wang et al. [44]. As described in Section 4.2. SYAUP-CN-26 was added to the above suspensions. The final concentrations were 1.823 mg/L, 19.263 mg/L, and 79.754 mg/L. The mycelia from the suspension were collected after 24 h in the 1.823 mg/L, 19.263 mg/L, and 79.754 mg/L SYAUP-CN-26 suspensions with suction filtration using a Buchner funnel. The mycelia were weighed (1.0 g) and ground to powder with liquid nitrogen. Then, a mixture (4 mL) containing 0.6% thiobarbituric acid and 10% trichloroacetic acid (dissolved in thiobarbituric acid with 1 M NaOH) was added. The mycelium suspensions were heated in boiling water (100 °C) for 15 min and cooled immediately with ice. Then, the mycelium suspensions were centrifuged at 10,000 *g* for 20 min to collect the supernatants. The malondialdehyde contents were determined using a UV spectrophotometer (UV1101M046, Varian, Australia). The detected wavelengths were 450 nm, 532 nm, and 600 nm. The MDA concentration was calculated using the following equations: c = 6.45 (OD_532_-OD_600_)-0.56 OD_450_. Each experiment was repeated three times.

### 4.5. Measurement of Reducing Sugar

The reducing sugar content was analyzed by the 3,5-dinitrosalicylic (DNS) colorimetric method [45]. As described in Section 4.2., SYAUP-CN-26 was added to the above suspensions. The final concentrations were 1.823 mg/L, 19.263 mg/L, and 79.754 mg/L. The treated suspensions were incubated (25 °C, 120 r/min) for 2, 6, 12, and 24 h. Reducing sugar was determined and calculated using an assay kit (Solarbio, Beijing). Each experiment was repeated three times.

### 4.6. Fluorescence Microscopy Observations

The membrane integrity of *B. cinerea* was determined according to Qin’s method [46] with some modifications. The suspensions of *B. cinerea* (5 × 10^5^ cfu/mL) were treated with 1.823 mg/L and 19.263 mg/L SYAUP-CN-26 in PDB medium and incubated (25 °C, 120 r/min) for 6 h. The spores in the PDB medium were collected by centrifugation at 5000 rpm for 10 min at 25 ± 1 °C, washed twice with sodium phosphate buffer (pH = 7.0), and centrifuged at 12,000 rpm for 2 min. After suspension, the spores were stained with 10 mg/L propidium iodide (PI) for 5 min at 30 °C. Finally, the spores were collected by centrifugation, washed twice with the buffer to remove any residual dye, and observed under a light microscope with an epifluorescence system (Eclipse E-200).

### 4.7. Scanning Electron Microscopy (SEM) Observations

The suspensions treated with 0 mg/L, 1.823 mg/L, and 19.263 mg/L SYAUP-CN-26 were incubated (25 °C, 120 r/min) for 48 h to obtain the mycelia. The spore suspensions (5 × 10^5^ cfu/mL) were cultured (25 °C) for 6 h with 0 mg/L, 1.823 mg/L, and 19.263 mg/L SYAUP-CN-26 for spore observations. The samples were treated with 2.5% glutaraldehyde for 3 h. Then, the samples were washed with sterile distilled water and transferred to a series of ethanol solutions (30%, 50%, 70%, and 90%) for 15 min each, followed by 100% ethanol for 20 min. After that, samples were air-dried, mounted on aluminum stubs, and sputter-coated with gold. The ultrastructure of the samples was observed using a BCPCAS4800 scanning electronic microscope (Hitachi, Tokyo, Japan) [47]. Each experiment was repeated three times.

### 4.8. Transmission Electron Microscopy (TEM) Observations

As described in Section 4.7, the mycelia was obstained. Then, they were treated with 2.5% glutaraldehyde at 4 °C for 3 h before being washed with sterile distilled water. Afterwards, samples were transferred to a series of ethanol solutions (30%, 50%, 70%, and 90%) for 1 h, respectively, followed by 100% ethanol for 2 h. After dehydrating and embedding in Spurr’s resin, ultrathin sections (< 100 nm) were cut with a diamond knife, collected on copper 300 mesh grids and allowed to dry on Formvar. The slices were then quickly double stained with uranyl acetate and lead citrate. Finally, the slices were prepared and visualized using a FEI Tecnai G20 Twin transmission electron microscope (FEI, Hillsboro, OR. USA) [47]. Each experiment was repeated three times.

### 4.9. Statistical Analysis

All values are expressed as means of the standard error of the mean. The statistical significance threshold (*p* < 0.05 for all analyses) was assessed by one-way ANOVA followed by Tukey’s post-hoc test for multiple comparisons using SPSS 21.0 software (SSPS Inc, Chicago, IL. USA).

## Figures and Tables

**Figure 1 molecules-25-00094-f001:**
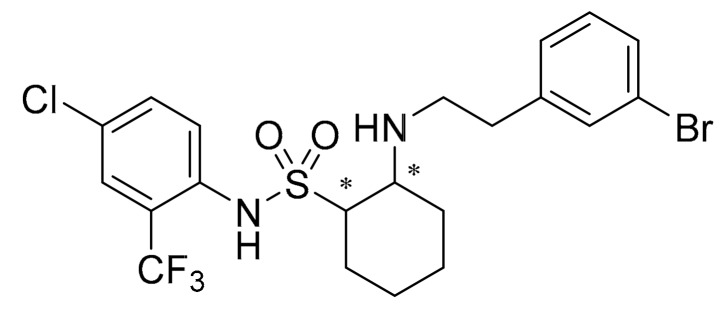
The structure of (1*S*,2*R*-((3-bromophenethyl)amino)-*N*-(4-chloro-2-trifluoromethylphenyl) cyclohexane-1-sulfonamide) (abbreviation: SYAUP-CN-26). * Indicates chiral carbon atoms.

**Figure 2 molecules-25-00094-f002:**
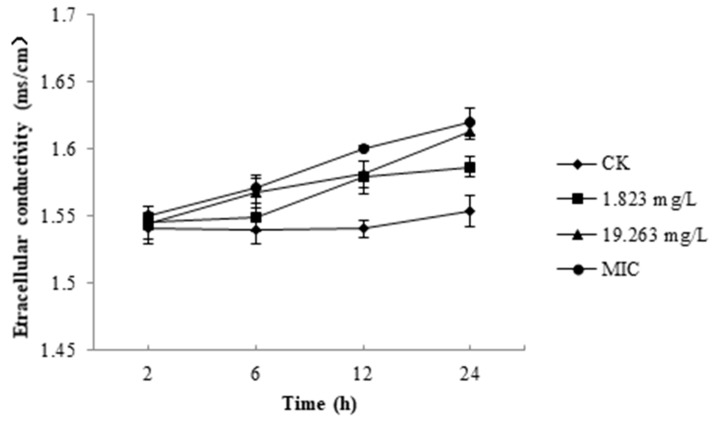
Effects of SYAUP-CN-26 on the relative electric conductivity of *Botrytis cinerea.* Values are presented as mean ± S.E. (*n* = 3) and are significant for * *p* < 0.05.

**Figure 3 molecules-25-00094-f003:**
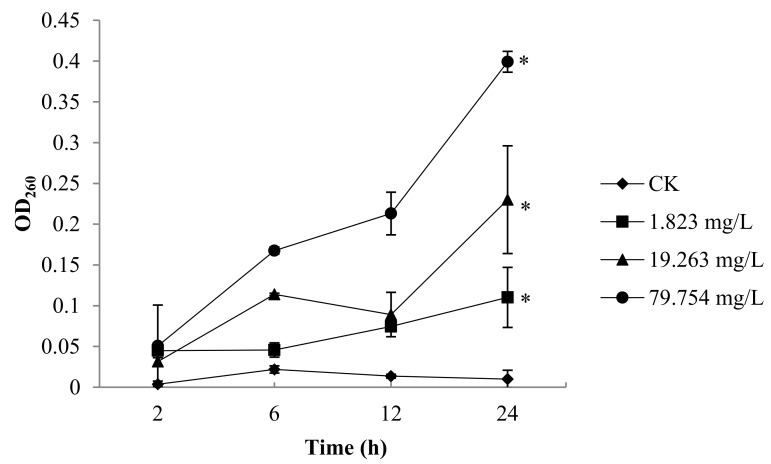
Effects of SYAUP-CN-26 on the nucleic acids of *B. cinerea.* Values are presented as mean ± S.E. (*n* = 3) and are significant for * *p* < 0.05.

**Figure 4 molecules-25-00094-f004:**
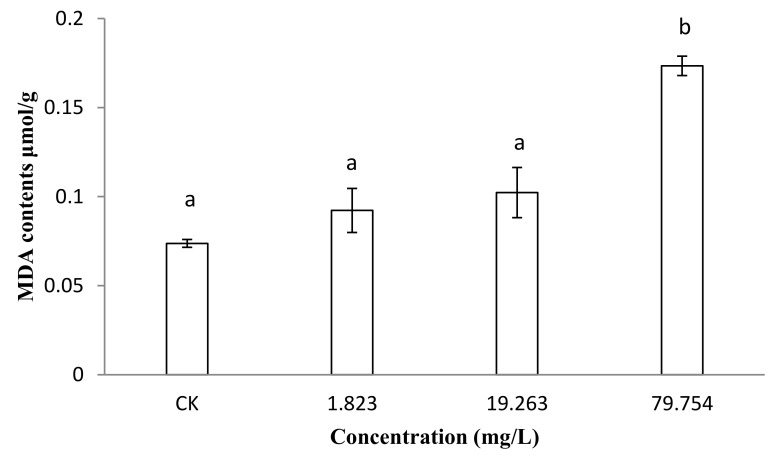
Effects of SYAUP-CN-26 on the malondialdehyde content (MDA) of *B. cinerea.* Values are presented as mean ± S.E. (*n* = 3). The lowercase letters at different concentrations indicate significant differences (*p* < 0.05).

**Figure 5 molecules-25-00094-f005:**
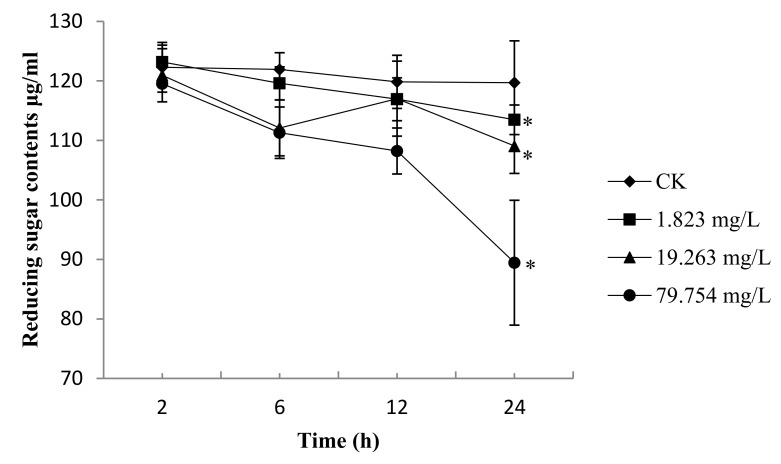
Effects of SYAUP-CN-26 on the reduction of reducing sugars content of *B. cinerea*. Values are presented as mean ± S.E. (*n* = 3) and are significant for * *p* < 0.05.

**Figure 6 molecules-25-00094-f006:**
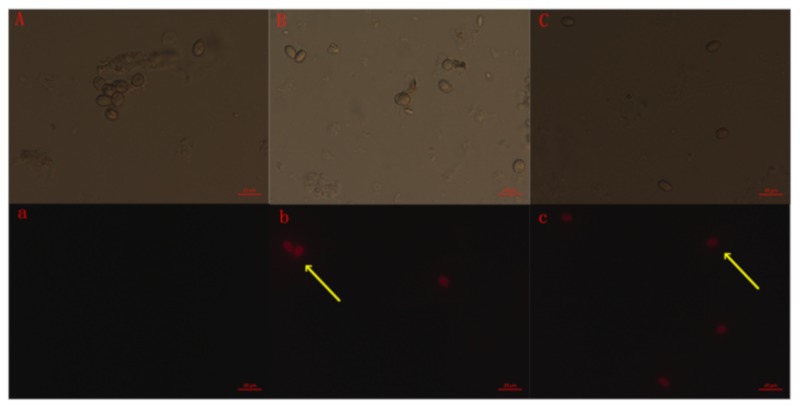
Effect of SYAUP-CN-26 on membrane permeability of *B. cinerea* spores. (**A**,**a**—0 mg/L; **B**,**b**— 1.823 mg/L; **C**,**c**—19.263 mg/L; **A**,**B**,**C**—*B. cinerea* spores under normal light microscope; **a**,**b**,**c**—under fluorescence microscope).

**Figure 7 molecules-25-00094-f007:**
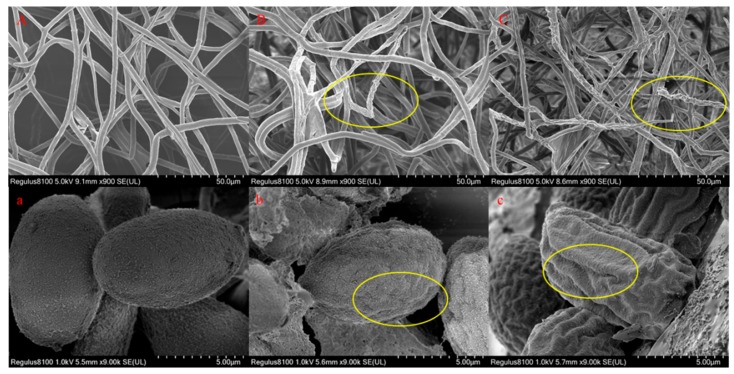
Scanning electron microscopy (SEM) analysis of the morphology of *B. cinerea* treated with SYAUP-CN-26. (**A**,**a**—0 mg/L; **B**,**b**—1.823 mg/L; **C**,**c**—19.263 mg/L; **A**,**B**,**C**—*B. cinerea* mycelia; **a**,**b**,**c**—*B. cinerea* spores. Yellow ovals indicate changes in the surface of mycelia and spores).

**Figure 8 molecules-25-00094-f008:**
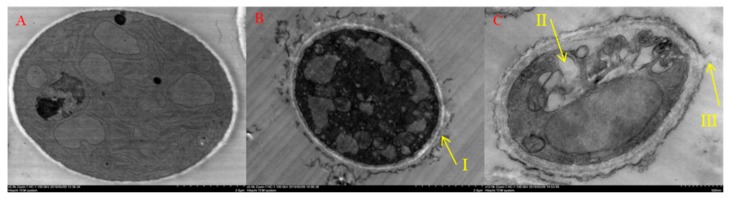
Transmission electron microscopy (TEM) analysis of the morphology of *B. cinerea* treated with SYAUP-CN-26. (**A**—0 mg/L; **B**—1.823 mg/L; **C**—19.263 mg/L; I and III indicate the cell membrane; II indicates the cytoplasm).

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
