# Peer review of "The Effect of (1S,2R-((3-bromophenethyl)amino)-N-(4-chloro-2-trifluoromethylphenyl) cyclohexane-1-sulfonamide) on Botrytis cinerea through the Membrane Damage Mechanism"

_molecules, 2019, doi:10.3390/molecules25010094_

Round 1

Reviewer 1 Report

Paper report the effect of new sulfonamide compound (SYAUP-CN-26) on electric conductivity, leakage of

nucleic acids, content of malondialdehyde, reducing sugars and membrane structure of B. cinerea. Overall, those results indicated that the compound could inhibit the growth of B. cinerea cells by damage cell membrane.

This paper has a discrete scientific interest, but it needs a careful revision with regard to the biological part. In detail:

1) page 2 line 54-55, the authors should specify studies already done on the molecule, specifying the references and the biological tests already carried out both on this molecule, but also  on similar molecules.

2) page 2, line 67/68, sentence not clear.  Authors should better specify.

3) Experimental part: the authors should specify which test is referred to MIC concentration ad clarify  te exact value. In addition authors should be specify why they used in all tests the other 2 concentrations (1.823 and 19.263.

4) in the conclusions if possible the authors should highlight the differences between the mechanism of action of the compound and that of other similar compounds.

Minor revision:

page 1, line 6 abstract: please insert "compound" instead of "compounds": Figure 1: draw the chiral carbon atoms appropriately; Figure 6. Better clarify the fluorescence test in the legend.

Reviewer 2 Report

In this work authors seek to explain the mechanism of action of SYAUP-CN-26 that produces damages to B. cinerea. They have used various methodologies to achieve their goal. Their research and investigations are original and interesting to the field. However, the paper needs to be improved before being published. Although English is not my native language, I think that the manuscript should be revised. I’m sure if this paper had been written in their language, they would have done a better work. In general, I think that with the help of someone with better English skills, authors will do a wonderful job.

Abstract

The abstract summarizes the work properly with and appropriate length. Please, do not use abbreviations. Replace PI and TEM.

Introduction

I’m sorry but I cannot be more specific to help the authors. In my opinion, the introduction is not clear enough. A little more information about Botrytis cinerea should be included. The goal of the paper needs to be mentioned explicitly in this section.

Results

Graphics should include the statistic significant differences when they are present.

Please, include that MDA is Malondialdehyde (MDA) content in 2.3. It is important to mention that it is used to measure the lipid peroxidation.

The quality of Figure 6 is not good enough.

Discussion

I like the discussion. In my opinion, authors have demonstrated that SYAUP-CN-26 affects the membrane permeability.

In the last paragraph they include a final remark that is useful to understand the main aim of the paper. This idea should be use at the end of their introduction.

Materials and Methods

The fungal strain: The strain was isolated by the authors. In order to reproduce their work, should be this strain available from a culture collection?

In general, I think that authors should include more information to allow colleagues to reproduce the experiments.

Electric conductivity of B. cinerea: there is no information about the acetone control and blank control.

Section 4.4- Authors include a reference to previously published methodology. Are all the modifications that they had made included in the manuscript?

Round 2

Reviewer 1 Report

the major part of my suggestions has been made by the authors, so in this form paper could be accepted

Reviewer 2 Report

The manuscript has been modified according to my suggestions.